# Preventive behavior of Vietnamese people in response to the COVID-19 pandemic

**Nhan Phuc Thanh Nguyen**[1], **Tuyen Dinh Hoang**[2], **Vi Thao Tran**[1], **Cuc Thi Vu**[1], **Joseph Nelson Siewe Fodjo**[3], **Robert Colebunders**[3], **Michael P. Dunne**[1,4], **Thang Van Vo**[1,2]*

**1** Institute for Community Health Research, University of Medicine and Pharmacy, Hue University, Hue, Vietnam, **2** Faculty of Public Health, University of Medicine and Pharmacy, Hue University, Hue, Vietnam, **3** Global Health Institute, University of Antwerp, Antwerpen, Belgium, **4** Faculty of Health, Queensland University of Technology, Brisbane, Australia

* vovanthang147@hueuni.edu.vn

**Data Availability Statement:** All relevant data are within the manuscript and its Supporting Information files.

## Abstract

We sought to evaluate the adherence of Vietnamese adults to Coronavirus Disease 2019 (COVID-19) preventive measures, and gain insight into the effects of the epidemic on the daily lives of Vietnamese people. An online questionnaire was administered from March 31 to April 6, 2020. The questionnaire assessed personal preventive behavior (such as physical distancing, wearing a face mask, cough etiquette, regular handwashing and using an alcohol hand sanitizer, body temperature check, and disinfecting mobile phones) and community preventive behavior (such as avoiding meetings, large gatherings, going to the market, avoiding travel in a vehicle/bus with more than 10 persons, and not traveling outside of the local area during the lockdown). A total adherence score was calculated by summing the scores of the 9 personal and the 11 community prevention questions. In total, 2175 respondents completed the questionnaire; mean age: 31.4 ± 10.7; (range: 18–69); 66.9% were women; 54.2% were health professionals and 22.8% were medical students. The mean adherence scores for personal and community preventive measures were 7.23 ± 1.63 (range 1–9) and 9.57 ± 1.12 (range 1–11), respectively. Perceived adaptation of the community to lockdown (Beta (β) = 2.64, 95% Confidence Interval (CI) 1.25–4.03), fears/worries concerning one's health (β = 2.87, 95% CI 0.04–5.70), residing in large cities (β = 19.40, 95% CI 13.78–25.03), access to official COVID-19 information sources (β = 16.45, 95% CI 6.82–26.08), and working in healthcare/medical students (β = 22.53, 95% CI 16.00–29.07) were associated with a higher adherence score to anti-COVID instructions. In conclusion, this study confirmed a high degree of adherence to personal and community preventive behavior among Vietnamese people. Our findings are consistent with the epidemiology of COVID-19 in Vietnam, where there have been few infections and no recorded deaths up to the first week of July 2020.

## Introduction

Coronaviruses (CoVs) are a group of viruses which co-infect humans and other animals of the vertebrates. CoV infections affect humans, cattle, birds, bats and many other wild animals in

**Funding:** This study received financial support from the Institute for Community Health Research, University of Medicine and Pharmacy, Hue University, Vietnam and Global Health Institute, University of Antwerp, Belgium (R Colebunders received funding from VLIRUOS for establishing the ICPcovid website).

**Competing interests:** The authors declare that there is no conflict of interest.

the respiratory, gastrointestinal, liver and central nervous systems [1]. In December 2019, a Coronavirus disease 2019 (COVID-19) outbreak erupted in China and has been spreading on a global scale [2]. Due to transmission via large droplets, aerosol and fomites, the novel coronavirus (SARS-CoV-2) spread rapidly around the planet [3]. Preventive public health measures have been implemented to fight the pandemic. Although the strategies applied internationally are similar, the timeliness, scale, and assertiveness of implementation regimes have varied considerably [4].

In Vietnam, the first person with a COVID-19 infection was detected on January 23rd and as of May 5th, 2020, Vietnam had totaled 271 confirmed cases with zero deaths [5]. Currently Vietnam is among the countries with the lowest number of reported cases, which is remarkable given its population size (approx. 95 million people) and proximity to the epicenter. From the start of the outbreak the government of Vietnam implemented intensive control in the northern Vinh Phuc province (considered to be the local focus of the disease) using a strategy of rapid testing for early detection of sources of infection, assertive contact tracing, timely isolation and free clinical care for people with the infection. Community preventive efforts were implemented early and have been pervasive throughout the country. The government supported social distancing, self-isolation of vulnerable people, mandatory isolation of symptomatic people and those who test positive, focal environmental sanitization, frequent hand washing and wearing of face masks in all public spaces.

By February 25th, one month after the first case was recorded, all patients had successfully recovered and had been discharged from hospitals. After more than 20 days with no new case reported, the 17th positive case of COVID-19 was confirmed on March 6th. Another wave of the epidemic hit the country with cases being imported from Europe, the USA, and other countries. This led to an increase in domestic transmission of COVID-19, thus ushering in the second stage of the epidemic. Fortunately, the government and health agencies had pandemic preparedness and control plans in place following the fairly recent experience with fighting the Severe Acute Respiratory Syndrome (SARS), Swine Flu (A/H1N1pdm09 virus, also known as 'H1N1'), and Avian influenza (Avian influenza virus subtypes A). The Government implemented national measures restricting travel and suspended visas for foreigners entering Vietnam. On March 20th, community transmission was indicated when the 86th and 87th COVID-19 patients had no travel history and no apparent contact with COVID-19 patients [5]. To further prevent disease spread in the community, on March 31st the Prime Minister mandated urgent measures, including strict social distancing throughout the country for 15 days. Accordingly, all people were required to stay at home, only go out in case of necessity, and keep a minimum distance of at least 2 meters when moving outdoors; shut down all non-essential business activities and services, only allow essential services such as food distribution, non-elective medical procedures, pharmacies store and the fuel supply. In addition, gatherings of more than 2 people were prohibited [6].

The primary purpose of this study was to assess how well Vietnamese adults have adhered to these instructions because they are crucial in preventing the spread of the virus. We also sought to investigate the effects of the epidemic on the daily lives of Vietnamese people.

## Methodology

### Study setting and design

We conducted a descriptive cross-sectional online survey among Vietnamese residents, during which we received voluntary responses continuously for seven consecutive days (from March 31st to April 6th, 2020).

## Study procedures

Data were collected through an online survey initiated by the ICPcovid consortium (https://www.icpcovid.com/). A secure website was used to design and host a questionnaire, which was developed to investigate individual/community factors that may influence adherence to COVID-19 preventive measures (Fig 1).

The research team adapted the international questionnaire **t**o the local Vietnamese context, translated it from English to Vietnamese, pilot-tested it, and improved the final questionnaire before official use. It took about 10 minutes to complete the questionnaire. The web link to the online survey was disseminated via various social media platforms, and consenting volunteers submitted their information anonymously. The data became available immediately upon submission. The online questionnaire was kept open for one week (recruitment period) after which it was closed and inaccessible.

## Sample size and sampling

Sampling was done using the snow-ball approach: as more persons completed the online questionnaire, they were encouraged to share the survey web link to their contacts. We opted for a convenience sample, whereby all eligible entries recorded within the one-week survey period

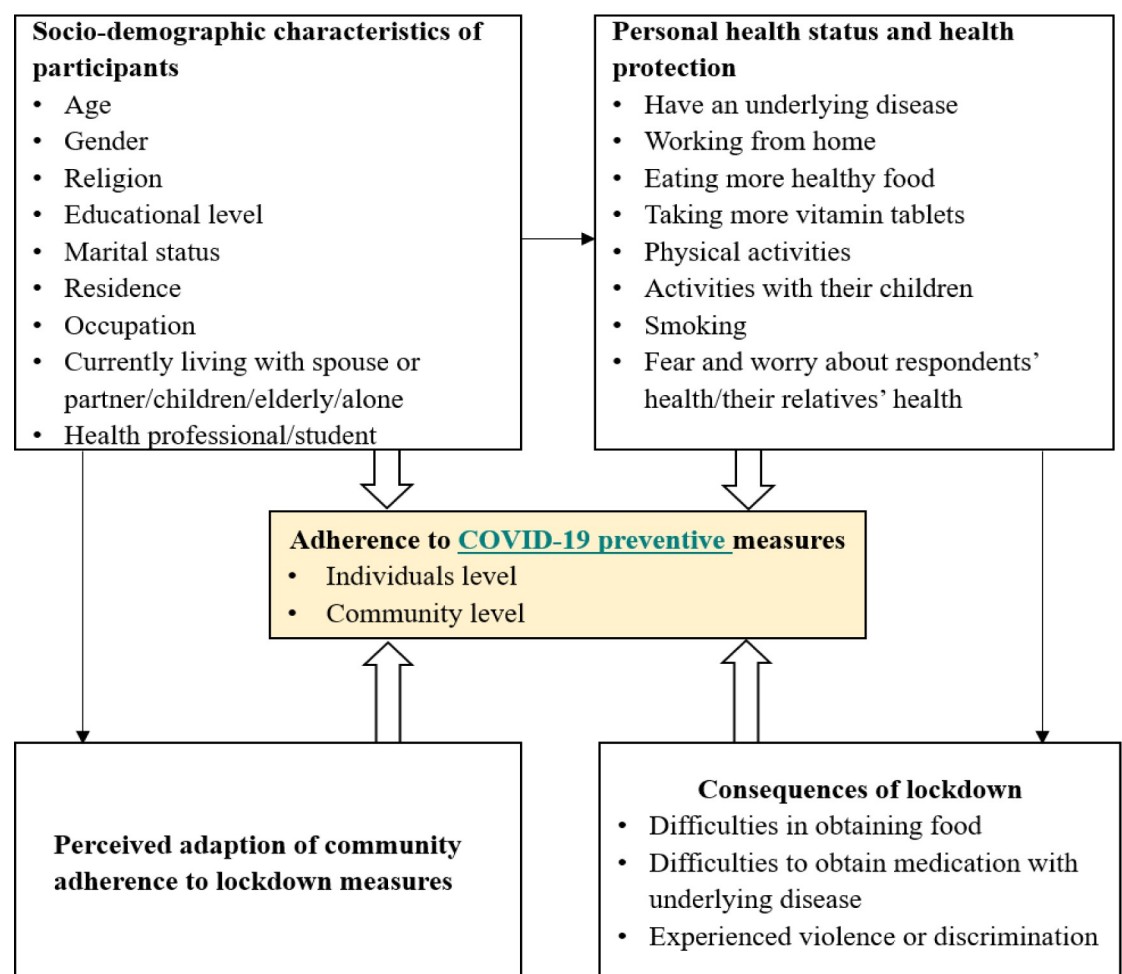

**Fig 1. Conceptual framework with the individual and community features investigated.**

were included in the study. Only data from respondents who self-identified as being at least 18 years old, who were Vietnamese citizens, understood the Vietnamese language, and resided in Vietnam at the time of the study were retained for analysis.

## Information collected from participants (see also Fig 1)

**Socio-demographic information.** The first part of the questionnaire gathered socio-demographic information, including participants' age, gender (male-female-other), profession (student, government staff, private enterprise, unemployed), urban vs rural residence, religion (no religion–religion), educational level (high school and lower–university and higher), marital status (married–not married), place of residence (municipalities–smaller urban or rural areas).

**Adherence to COVID-19 preventive measures.** Adherence to **personal preventive measures** was assessed by using 9 questions, covering the following aspects: following the 1.5-2m meters physical distance rule; wearing a face mask when going outside; avoiding touching the face; covering of mouth and nose when coughing/sneezing; hand hygiene via regular hand washing and/or disinfection with sanitizer; frequency of body temperature check; disinfecting mobile phone frequently. Additionally, we asked participants to self-evaluate how difficult it was for them to stay at home as required by the lockdown, and this was reported using a 5-point Likert scale (1 = not difficult at all, to 5 = extremely difficult). Adherence to **community preventive measures** was assessed with 11 questions with a focus on the following strategies: avoiding meetings/gatherings; avoiding being in a vehicle/bus with more than 10 persons; avoiding going to crowded entertainment venues/ public gym/ beauty salon; avoiding funeral attendance; avoiding going to a fresh food market; usage of individual spoons and plates when eating together with family/non-family members; avoiding traveling to another province/ country during the lockdown period.

**Information about daily life during the COVID-19 lockdown.** Additional questions were asked about daily life and professional activities during the COVID-19 pandemic. Fears about the participants' health as well as their family well-being were measured on a 5-point Likert scale (1 = not worried/afraid, to 5 = extremely worried/afraid). For both variables, a score of 3 or above was considered as moderate to high level of fear/worry. Possible difficulties with daily life activities during the previous week (such as working from home, access to food, access to medication for respondents with underlying chronic conditions, violence/discrimination as a consequence of the lockdown measures) were also assessed using yes/no questions.

Similarly, the degree of adaptation of the community to lockdown instructed from the government, as perceived by respondents, was evaluated using a 10-point Likert scale (1 = no adaption, to 10 = very strong adaptation). A score $\geq 6$ on the 10-point Likert scale was considered as good adaptation to the government's instructions. Respondents were also asked about their sources of COVID-19 information; possible responses were: "National television", "Radio", "Vietnamese Ministry of Health website", and "WHO website" were all considered as official sources, while other sources (including social media) were considered unofficial.

## Ethical considerations

Anonymity and informed consent were assured. The study was approved by the Ethical Review Committee of Hue University of Medicine and Pharmacy, Vietnam (No. H202/041 dated March 30th, 2020).

## Data analysis

Statistical analysis was performed using SPSS version 20.0. Descriptive statistics presented continuous data as mean ± standard deviation (SD), while categorical variables were presented as

percentages. Given that we sought to identify predictors of adherence to COVID-19 preventive measures, a multiple linear regression model was used to analyze which independent variables associated with squared-transformed adherence scores. Model covariates included age and gender, as well as other variables which showed a significant association with the dependent variable during bivariate analysis. 95% confidence intervals and a p-value of less than 0.05 were used for significance testing.

## Results

### Respondent characteristics and their daily activities

A total of 2192 persons completed the online questionnaire. After data cleaning and application of inclusion criteria, 2175 responses were kept. The participants resided in 55/63 provinces of Vietnam: 1054 (48.5%) lived in major municipalities (Ha No, Ho Chi Minh City, Hai Phong, Can Tho, and Da Nang) and 1121 (51.5%) lived in smaller urban or rural areas. The mean age was 31.39 years (SD: 10.66, range: 18–69), and the majority of participants (66.9%) were women. The characteristics of our study participants are summarized in Table 1.

**Table 1. Characteristics of study participants (n = 2175).**

| Characteristic | | n | % |
|---|---|---|---|
| Gender | Male | 716 | 32.9 |
| | Female | 1454 | 66.9 |
| | Other | 5 | 0.2 |
| Highest educational level | High school and lower | 496 | 22.8 |
| | University and higher | 1679 | 77.2 |
| Marital status | Married | 1000 | 46.0 |
| | Not married | 1175 | 54.0 |
| Religion | Has a religion | 524 | 24.1 |
| | No religion | 1651 | 75.9 |
| Place of residence | Municipalities | 1054 | 48.5 |
| | Smaller urban or rural areas | 1121 | 51.5 |
| Occupation | Student | 542 | 24.9 |
| | Government staff | 768 | 35.3 |
| | Private enterprise | 766 | 35.2 |
| | Unemployed | 99 | 4.6 |
| Professional | Health professional | 1178 | 54.2 |
| | Medical student | 495 | 22.8 |
| | Non-health professional/student | 502 | 23.0 |
| Urban/Rural or Semi-Rural residence | Urban | 1431 | 65.8 |
| | Sub-urban/Rural | 744 | 34.2 |
| Currently living with: | Alone | 139 | 6.4 |
| | With children | 1232 | 56.6 |
| | With the elderly | 332 | 15.3 |
| | Spouse or partner | 929 | 42.7 |
| Currently smoking | Yes | 147 | 6.8 |
| | No | 2028 | 93.2 |
| Eating more healthy food since the COVID epidemic | Yes | 1917 | 88.1 |
| | No | 258 | 11.9 |
| Taking more vitamin tablets since the COVID-19 epidemic | Yes | 1262 | 58.0 |
| | No | 913 | 42.0 |

## Impact of COVID-19 on respondents' domestic and professional habits

Most participants said they obtained COVID-19 information through official sources such as state television (81.1%) and the Ministry of Health of Vietnam website (74.5%). Of the 1613 participants with a stable job, 777 (48.2%) started working from home because of the epidemic. Confinement measures resulted in 133 (6.1%) participants experiencing difficulties in obtaining food, and 42 (1.9%) persons reported suffering from some form of violence/discrimination because of the restrictive measures taken against COVID-19. Moreover, on a 5-point Likert scale, 30.0% and 42.3% of respondents reported that they had moderate to high levels of fear and worry about their own health, and that of their relatives, respectively (Table 2).

## Adherence to preventive measures in the national response to the threat of COVID-19

**Adherence to personal preventive measures.** "Wearing a face mask when going outside" had the highest adherence rate of 99.5%. Adherence to "regular hand washing using soap and water" and "covering of mouth and nose with a tissue paper was also high with rates reaching 97.4% and 94.9% respectively. However, adherence regarding "temperature measurement at least twice a week" was low at 45.1% (Table 3). Using a 9-item score, the mean level of personal adherence to preventive measures was 7.23 ± 1.63; range: 1 to 9. At the individual level, participants reported a low level of difficulty in complying with the stay-at-home measures (mean difficulty score on the Likert scale: 1.69 ± 0.86; range 1 to 5).

**Adherence to community preventive measures.** During the week preceding the survey, most of the participants responded that they "Had not traveled to another province/country", "Avoided going to a religious gathering", and "Avoided going to a public gym" with adherence rates at 99.4%, 99.3% and 99.2%, respectively. However, nearly half of the participants had visited a fresh food market in the past seven days (Table 4). Adherence scores for community preventive measures, as assessed by 11 questions, ranged from 0 to 11; mean score: 9.57 ± 1.12. A majority of respondents (76%) reported moderate to high adaptation of their community members in compliance with the government's instructions.

**Table 2. Impact of COVID-19 confinement measures on domestic and professional habits.**

| Characteristic | | n | % |
|---|---|---|---|
| Fear and worry about respondents' health | Yes | 652 | 30.0 |
| | No | 1523 | 70.0 |
| Fear and worry about their relatives' health | Yes | 919 | 42.3 |
| | No | 1256 | 57.7 |
| Difficulties in obtaining food | Yes | 133 | 6.1 |
| | No | 2042 | 93.9 |
| Working from home (n = 1613) | Yes | 777 | 48.2 |
| | No | 836 | 51.8 |
| Experienced violence or discrimination during the confinement | Yes | 42 | 1.9 |
| | No | 2133 | 98.1 |
| Physical activity during the epidemic (n = 453) | Yes | 421 | 92.9 |
| | No | 32 | 7.1 |
| Type of physical activity | Indoor, with music | 148 | 32.7 |
| | Indoor, with online video | 49 | 10.8 |
| | Outdoor | 271 | 59.8 |

**Table 3. Adherence to personal preventive measures for COVID-19.**

| N° | Characteristics | n | % |
|---|---|---|---|
| 1. | Follow the 1.5-physical distance rule | 1919 | 88.2 |
| 2. | Face mask use when outdoor | 2165 | 99.5 |
| 3. | Cover mouth and nose when coughing/sneezing | 2065 | 94.9 |
| 4. | Usually wash/disinfect hands immediately after coughing/sneezing | 1813 | 83.4 |
| 5. | Wash hands regularly with water and soap during the day | 2119 | 97.4 |
| 6. | Use hand sanitizer/gel regularly during the day | 1767 | 81.2 |
| 7. | Body temperature check at least twice a week | 980 | 45.1 |
| 8. | Avoid touching my face, eyes, nose and mouth with my hands | 1852 | 85.1 |
| 9. | Disinfect phone when I get home | 1047 | 48.1 |

## Factors associated with adherence to government measures against COVID-19

Summing the responses from self-reported adherence to both personal and community prevention strategies, we produced an overall adherence score (Total score: 9+11 = 20). Respondents' scores ranged from 2 to 20, with a mean of 16.80 ± 2.13. Total adherence scores were square-transformed to approximate a normal distribution and used as the dependent variable in linear regression models investigating factors associated with adherence to preventive measures (Table 5). We observed that worries about one's health (β = 2.87, p = 0.047), perceived adaptation of the community to the lockdown (β = 2.64, p<0.001), residence in large municipalities (β = 19.40, p<0.001), official sources of Covid-19 information (β = 16.45, p = 0.001), and having a professional role in the health sector (worker or student) (β = 22.53, p<0.001) were associated with higher adherence scores. Conversely, people who reported higher perceived difficulty in obeying lockdown instructions (β = -23.97, p<0.001) had significantly lower adherence scores after adjusting for socio-demographic characteristics and other confounders (Model adjusted R-squared = 0.144).

## Discussion

The government of Vietnam took relatively prompt and intensive measures to reduce the spread of COVID-19 infection in Vietnam. Our data show that most Vietnamese people who participated in the survey complied with most strategies to prevent infection. Very few people

**Table 4. Adherence to community preventive measures for COVID-19.**

| N° | Characteristics | n | % |
|---|---|---|---|
| 1. | Avoided meeting or gathering with more than 10 persons in last seven days | 1791 | 82.3 |
| 2. | Avoided going to a restaurant, bar, or club in the last seven days | 2147 | 98.7 |
| 3. | Avoided attending a funeral in the last seven days | 2117 | 97.3 |
| 4. | Avoided going to a religious gathering during the last seven days | 2160 | 99.3 |
| 5. | Avoided going to a public gym in the past 7 days | 2157 | 99.2 |
| 6. | Avoided going to a beauty parlor, massages, spa, hairdresser or nail studio | 2121 | 97.5 |
| 7. | Avoided being in a vehicle or bus with more than 5 persons in last seven days | 2079 | 95.6 |
| 8. | Avoided using of common plates/spoons when eating with family during last seven days | 1137 | 52.3 |
| 9. | Avoid using of common plates/spoons when eating with strangers during last seven days | 1986 | 91.3 |
| 10. | Avoided going to a market during the last seven days | 950 | 43.7 |
| 11. | Had not traveled outside my city during the last seven days | 2162 | 99.4 |

**Table 5. Multiple linear regression investigating factors associated with adherence to the COVID-19 preventive measures.**

| Co-variates | Estimate (95% Confidence interval) | P-value |
|---|---|---|
| Age | -0.08 (-0.36–0.20) | 0.567 |
| Gender: Male | 2.86 (-3.09–8.81) | 0.346 |
| Fear and worry about their own health (Likert score) | 2.87 (0.04–5.70) | 0.047 |
| Perceived adaptation of the community to lockdown (Likert score) | 2.64 (1.25–4.03) | <0.001 |
| Difficulty in obeying lockdown (Likert score) | -23.97 (-27.39 – -20.55) | <0.001 |
| Residence in large Municipalities | 19.40 (13.78–25.03) | <0.001 |
| Official sources to obtain Covid-19 information | 16.45 (6.82–26.08) | 0.001 |
| Being a healthcare worker/student | 22.53 (16.00–29.07) | <0.001 |

resisted the orders for using face masks, frequent hand washing, avoiding large gatherings, or proper social distancing. This high uptake of protective behaviors is consistent with the epidemiological trends for COVID-19 in Vietnam. The spread of the infection has been minimized, following full implementation of prevention strategies for the whole population such that between April 16[th] and the first week of July 2020, there have been no new COVID-19 cases resulting from community transmission [7].

Most companies and state organizations have implemented unprecedented working methods in accordance with national efforts to promote working from home where it is feasible. This study found that 48.2% of workers were obliged to work from home during the COVID-19 confinement. Although negative effects of lockdown on people's jobs and lives might emerge if sustained for long periods, the participants in this study indicated a relatively low level of difficulty to stay at home in the short term. The most frequently reported difficulty encountered during the lockdown had to do with meeting daily needs for food.

## Respondents' adherence to COVID-19 preventive measures

The survey revealed that although 30.0% of respondents were moderately to severely worried/afraid about their own health, a greater proportion (42.3%) was concerned about the health risks for family members. This may reflect the mean age of participants; as most were young adults, they may be concerned about risks to older family members which is particularly relevant in Vietnam where many people live in multi-generational extended family households.

It is common and easy to apply measures such as wearing a mask and washing hands frequently with soap or disinfectant solutions. Although the efficacy of non-medical masks in preventing COVID-19 spread is currently subject to debate, mask use among infected persons can limit the spread of the virus to the outside environment [8–10]. The rate of wearing masks when going out in this study was 99.5%, similar to an estimate of 98% in a Chinese study but higher than 70.1% observed in Japan [10, 11]. Two reasons for such high mask use are the fact that the Vietnamese government made mask use mandatory from April 1[st], and that in many parts of the country, a majority of the people have a habit of wearing masks to cope with air pollution [6, 12]. Although negative social interactions regarding face mask usage have been reported in some parts of the World [13], in Vietnam and some East Asian countries such as China, Japan, and Korea, wearing face masks is ubiquitous [14]. It has been practiced for health and cultural reasons [8, 14], so the transition to more widespread mask wearing in response to COVID-19 appears not to have caused a conflict that can sometimes arise if people are forced to change cultural norms.

Community prevention measures were implemented very early in response to a localized outbreak in a northern province, and this was re-enforced from April 1[st] with official

implementation of nationwide lockdown. Such a national shutdown was unprecedented in Vietnam, with all except essential businesses closed [6]. People were advised to stay at home as a patriotic act, and only go out when necessary. Information about outbreaks in healthcare, religious gatherings and entertainment facilities was disseminated widely via mainstream and social media [15, 16]. In this survey, the item "Avoided going to a fresh market" had the lowest adherence (43.7%), probably because fresh foods are indispensable in the household and also due to the fact there were more women (66.9%) in the sample and women tend most often to procure fresh food in Vietnam. It is worth noting that in the national lockdown regulations, going to the market is a valid reason to leave the house, although people were asked to reduce the frequency of this activity to the bare minimum [6].

People living in Municipalities had higher adherence scores, perhaps because about 70% of the COVID-19 cases were diagnosed in cities [17]. Many respondents were working as health-care professionals or were medical students, so they may tend to be more adherent to health protection efforts. Age and gender were not significantly associated with adherence score in this study (Table 5). The high adherence to government recommendations has proven extremely important in the fight against COVID-19 infection. Good adherence to the preventive measures indicates that most people in the survey tend to support the Government's public health motives and requirements, showing patriotism, solidarity and rapid adoption of preventive behaviours during the epidemic. According to Berlin-based Dalia Research, 62% of respondents in Vietnam believe the government is doing the "right amount" in response to the COVID-19 pandemic [18]; it is therefore not surprising that Vietnam has been internationally recognized for their success in controlling COVID-19 [19]. In our study, the proportion of respondents receiving information from reliable sources was high, which suggests that most people were careful to avoid unreliable advice and deliberate misinformation. Notably, the Vietnamese government has sanctioned acts that spread fake news [20].

## Study limitations

There are several limitations of this study. First, the participants were not a representative sample of the Vietnamese population. Indeed, respondents were mainly people from medium to high social strata, since poor and vulnerable populations in Vietnam may have limited internet access. The snowball sampling method and medical university-based recruitment over just one week explains the fact that health professionals, health science students, and female respondents were over-represented. Random sampling of the population is necessary. Second, it is not possible to verify the veracity of responses provided via a web-based questionnaire. Third, the cross-sectional study design provided only a snapshot of preventive behaviour over one week. It will be important to monitor adherence to official recommendations over time as societies adapt to changing conditions throughout the unpredictable course of this pandemic.

## Conclusion

The study provides insight into compliance with the national lockdown and other risk mitigation measures implemented in Vietnam in the context of the COVID-19 pandemic. Overall, adherence to government instructions was high and most likely played a role in rapidly controlling the epidemic in Vietnam and limiting its public health impact. Since April 27th the strict lockdown measures were stopped and life is gradually returning to normal in Vietnam, albeit with a stronger than usual emphasis on personal protection during social interactions. Careful monitoring for potential new imported COVID-19 infections and community transmission is needed to prevent a resurgence of the epidemic.

## Supporting information

**S1 File.**
(DOCX)

## Acknowledgments

We are grateful to the respondents for their participation. The authors would also like to thank all institutions and stakeholders across the country for supporting us to collect data via online questionnaires. Finally, we would also like to acknowledge Prof. Nguyen Vu Quoc Huy, Rector of University of Medicine and Pharmacy, Hue University, Vietnam for his wonderful support for conducting this study in the difficult time of COVID-19 pandemic occurred.

## Author Contributions

**Conceptualization:** Nhan Phuc Thanh Nguyen, Joseph Nelson Siewe Fodjo, Robert Colebunders, Thang Van Vo.

**Formal analysis:** Nhan Phuc Thanh Nguyen, Tuyen Dinh Hoang.

**Investigation:** Vi Thao Tran, Cuc Thi Vu, Thang Van Vo.

**Methodology:** Nhan Phuc Thanh Nguyen, Tuyen Dinh Hoang, Vi Thao Tran, Cuc Thi Vu, Robert Colebunders, Thang Van Vo.

**Writing – original draft:** Nhan Phuc Thanh Nguyen, Tuyen Dinh Hoang, Vi Thao Tran, Cuc Thi Vu, Robert Colebunders.

**Writing – review & editing:** Nhan Phuc Thanh Nguyen, Joseph Nelson Siewe Fodjo, Robert Colebunders, Michael P. Dunne, Thang Van Vo.

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
