## [Decision Letter · Decision Letter 0]

9 Jul 2020

PONE-D-20-13904

Preventive behavior of Vietnamese people in response to the COVID-19 pandemic

PLOS ONE

Dear Dr. Vo,

Thank you for submitting your manuscript to PLOS ONE. After careful consideration, we feel that it has merit but does not fully meet PLOS ONE’s publication criteria as it currently stands. Therefore, we invite you to submit a revised version of the manuscript that addresses the points raised during the review process.

1. While you intended to gather a broad community sample, you obtained predominantly a sample of healthcare workers in Vietnam. Please re-analyze with just the healthcare workers and rephrase descriptors to indicate this it the population considered.

2. Avoid superlative descriptors, ex. "excellent preventive behavior."

We look forward to receiving your revised manuscript.

Kind regards,

James Curtis West, M.D.

Academic Editor

PLOS ONE

Journal Requirements:

2. Please include additional information regarding the survey or questionnaire used in the study and ensure that you have provided sufficient details that others could replicate the analyses. For instance, if you developed a questionnaire as part of this study and it is not under a copyright more restrictive than CC-BY, please include a copy, in both the original language and English, as Supporting Information. Moreover, please include more details on how the questionnaire was pre-tested, and whether it was validated.

3. In your Methods section, please provide additional information about the participant recruitment method and the demographic details of your participants. Please ensure you have provided sufficient details to replicate the analyses such as: a) the recruitment date range (month and year), b) a description of any inclusion/exclusion criteria that were applied to participant recruitment, c) a table of relevant demographic details, d) a statement as to whether your sample can be considered representative of a larger population, e) a description of how participants were recruited, and f) descriptions of where participants were recruited and where the research took place.

Additional Editor Comments (if provided):

Thank you for the opportunity to review this interesting manuscript. Your paper examines adherence with COVID-19 preventive measures in an online sample of Vietnamese participants. Overall this study has meaningful data that contributes to knowledge of COVID impact. Please consider the following in improving your manuscript.

1. Your abstract concludes with the statement that preventive behavior was "excellent" and then states that this behavior explains the low number of infections and mortality. The first statement presents the appearance of bias and should be rephrased. The implication that these behaviors led to low infection rates and zero mortality cannot be supported based on your evidence alone and should be removed.

2. In your sample, healthcare workers appear to be heavily over-represented. One could reasonably consider this to be a sample of healthcare worker behaviors rather than a general population sample. Please consider revising your analysis to only include healthcare workers. While not your original intent, this might be a more meaningful analysis.

3. In your discussion, you state that "positive attitudes" indicate most people believe in government requirements. Your data does not support this statement, as you only measured reported compliance and did not ask about attitudes toward the requirements. If information about attitudes toward requirements was obtained, it should be presented in the results section clearly.

Reviewers' comments:

Reviewer's Responses to Questions

**Comments to the Author**

1. Is the manuscript technically sound, and do the data support the conclusions?

Reviewer #1: No

Reviewer #2: Yes

2. Has the statistical analysis been performed appropriately and rigorously? 

Reviewer #1: Yes

Reviewer #2: Yes

3. Have the authors made all data underlying the findings in their manuscript fully available?

Reviewer #1: No

Reviewer #2: Yes

4. Is the manuscript presented in an intelligible fashion and written in standard English?

Reviewer #1: Yes

Reviewer #2: Yes

5. Review Comments to the Author

Reviewer #1: Nguyen et al conducted an online questionnaire survey to evaluate the adherence of Vietnamese adults to COVID-19 preventive measures. The topic is interesting. However, there are several methodological issues which lead to large limitations of this study.

1. Although the participants resided in 55/63 provinces of Vietnam, there were only about 40 participants in a province. The sample size was not estimated. It is not sure whether the sampling is representative.

2. Due to the snow-ball sampling, there were apparently unbalanced characteristics of study participants investigated, such as the number of female was larger than that of male. These might affect the results of the study.

3. The author should provide the questionnaire as a supplementary file to show more details of the survey.

4. The definitions of some abbreviations or parameters should be given at their first used, such as CI and β in the Abstract section.

Reviewer #2: This well-written, interesting manuscript examines the factors associated with adherence to personal and community measures during the week in which strict mitigation efforts were established in Vietnam to reduce the spread of COVID-19. The study provides an informative snapshot of community members' behaviors during that period, and considers factors that may influence adherence to preventive measures. These findings have significant public health implications in the event of a subsequent wave of COVID-19. There are areas in which additional clarification would benefit the reader, including details regarding assessment. Further, the sample is limited by its high proportion of health professionals/medical students, with associated higher education level and socioeconomic status. Although this is considered in the Discussion, it is noteworthy and may influence the generalizability of the findings. I have commented on areas to address below.

Abstract

1. Page 2, line 22: Please spell out Coronavirus Disease 2019 (COVID-19) in its first use.

2. Page 2, line 24: Please consider editing as "...survey was administered..." or "...data were collected via an online questionnaire..." or something similar (versus organized)

3. Page 2, line 25: Please briefly describe the preventative behaviors referenced here.

4. Page 2, line 26: Please provide the mean (SD) age.

5. Page 2, line 27: How many items were included, and how were these items scored? This is needed to be able to interpret these scores, and put these findings in context. Please indicate whether adherence to personal and community measures were considered together in analyses.

6. Page 2, line 34: Please identify this low number based on the Vietnamese population.

Introduction

7. Page 3, line 38: Please begin by identifying the virus SARS-CoV-2.

8. Page 3, line 50: Were there any mandatory stay-at-home/quarantine orders in effect at this time, or was isolation at this point strictly among those who were infected?

9. Page 4, line 61: Please provide the scientific names of these infectious outbreaks.

10. Page 4, line 74: Please specify what types of effects were assessed (changes in routine activities, behavioral, psychological, physical)?

Methods

11. Page 5, line 85: How was consent indicated on this online assessment? Were the participants compensated in any way?

12. Page 5, lines 97-98: Please provide details of the categories of these socio-demographic variables.

13, Page 5, line 103: Please specify that this is adherence to personal and community measures.

14. Page 6, line 122: This is somewhat unclear- how is perceived adaptation of the community defined/conceptualized in this context?

15. Page 6, lines 119-124: Were these relationships to adherence still examined or strictly controlled for as covariates? These are important factors that could have a significant impact on adherence. Please clarify their role in analyses.

16. Page 7, line 133: Please clarify whether this was one multiple linear regression or a series of models, and specify how covariates were included. Also, please identify all of the independent variables here.

Results

17. Page 7, lines 140-141: How is there overlap, as these percentages exceed 100%? Perhaps identifying the specific categories and their associated percentages would be helpful. Also, these percentages do not correspond with what is included in Table 1 on page 8.

18. Page 7, Table 1: As mentioned earlier, please include all categories. Also, please reformat the table so that the categories align better with the descriptors/category levels in the middle columns.

19. Page 8, Table 1, Health Professional/Student category: Why were health professional and student combined? It is also not specified that these are medical students. This is a very high proportion of the total sample. Please provide explanation regarding whether sampling was targeted to health care professionals, or was this a community sample? Also, there is a possibility that health professional and medical students, despite their training, may have different experiences in their response to the pandemic. Please consider conducting the analyses with these groups examined separately.

20. Page 8, Table 1, Currently living with: Is living with spouse or partner a category? Again, please include all categories of all of the variables included in the analyses.

21. Page 8, Table 1, Smoking/Eating more healthy food/Taking more vitamin tablets: Is this within the past week? Please specify.

22. Page 8, line 152: These items need to be specified within the Measures section. It is not until this point that the reader is aware that these factors were assessed. Please confirm that all variables in the study are defined/described in the Measures section.

23. Page 8, line 154: Please initially report this coding scheme in the Methods/Measures section.

24. Page 9, Table 2: As with Table 1, please reformat, as it is unclear how these variables are categorized, and the alignment seems off. Please address with all tables.

25. Page 9, line 163: Please identify the percentage of those who responded 4 or 5 (or moderate or more?). Please ensure that all of the response options are listed in the Measures section.

26. Page 9, line 178: This categorization should be initially introduced int he Measures section.

27. Page 11: If this corresponds with the journal's standards, please include the B and p-value in text as well, to provide the reader with information about the direction and strength of the associations.

28. Page 190: Please provide more details regarding the "official sources of COVID-19 information" factor. All of the factors need to be clearly defined/identified in the Measures section, as the study examines many variables.

Discussion

29. Page 13, line 215: Please include all of these rating categories in the Measures section.

30. Page 13, line 219: Please add 'and' ("...such as wearing a mask and washing hands frequently...")

31. Page 14, bottom of page: Did the authors examine whether there were other demographics associated with adherence?

32. Page 15: Please provide additional interpretation of the main findings of the study (worries about health, difficulty staying at home).

33. Page 15, line 262: Can the authors identify the proportion of those of lower socioeconomic status having internet access in Vietnam, or more details about this in general? It is also important to comment on the possibility that certain demographic groups may not have comparable access to means by which to stay safe (social distancing, ready access to clean or disposable masks and hand sanitizer).

6. PLOS authors have the option to publish the peer review history of their article (what does this mean?). If published, this will include your full peer review and any attached files.

Reviewer #1: No

Reviewer #2: No

---

## [Author Response · Author response to Decision Letter 0]

31 Jul 2020

On behalf of our authoring team members, we thank you very much for your very useful comments to have our manuscript qualified better at the Plos One Journal's requirement. We do hope to continue getting your further reviews if our quality of revised manuscript is still scientifically limited.

---

## [Editor Report · Decision Letter 1]

26 Aug 2020

Preventive behavior of Vietnamese people in response to the COVID-19 pandemic

PONE-D-20-13904R1

Dear Dr. Vo,

We’re pleased to inform you that your manuscript has been judged scientifically suitable for publication and will be formally accepted for publication once it meets all outstanding technical requirements.

Kind regards,

James Curtis West, M.D.

Academic Editor

PLOS ONE

Additional Editor Comments (optional):

Thank you for your thoughtful consideration of the reviewer comments. This is a significantly improved manuscript.
---

## [Editor Report · Acceptance letter]

31 Aug 2020

PONE-D-20-13904R1 

Preventive behavior of Vietnamese people in response to the COVID-19 pandemic 

Dear Dr. Vo:

I'm pleased to inform you that your manuscript has been deemed suitable for publication in PLOS ONE. Congratulations! Your manuscript is now with our production department. 

Kind regards, 

on behalf of

Dr. James Curtis West 

Academic Editor

PLOS ONE